# Human Motion Pattern Recognition and Feature Extraction: An Approach Using Multi-Information Fusion

**DOI:** 10.3390/mi13081205

**Published:** 2022-07-29

**Authors:** Xin Li, Jinkang Liu, Yijing Huang, Donghao Wang, Yang Miao

**Affiliations:** 1School of Mechanical and Materials Engineering, North China University of Technology, Beijing 100144, China; ncut_liujinkang@126.com (J.L.); ncut_huangyijing@163.com (Y.H.); ncut_wangdonghao@126.com (D.W.); 2Faculty of Materials and Manufacturing, Beijing University of Technology, Beijing 100124, China; miaoyang@vip.126.com; 3Beijing Key Laboratory of Advanced Manufacturing Technology, Beijing University of Technology, Beijing 100124, China

**Keywords:** feature extraction, artificial intelligence, motion pattern recognition, wearable sensors, EMG, IMU

## Abstract

An exoskeleton is a kind of intelligent wearable device with bioelectronics and biomechanics. To realize its effective assistance to the human body, an exoskeleton needs to recognize the real time movement pattern of the human body in order to make corresponding movements at the right time. However, it is of great difficulty for an exoskeleton to fully identify human motion patterns, which are mainly manifested as incomplete acquisition of lower limb motion information, poor feature extraction ability, and complicated steps. Aiming at the above consideration, the motion mechanisms of human lower limbs have been analyzed in this paper, and a set of wearable bioelectronics devices are introduced based on an electromyography (EMG) sensor and inertial measurement unit (IMU), which help to obtain biological and kinematic information of the lower limb. Then, the Dual Stream convolutional neural network (CNN)-ReliefF was presented to extract features from the fusion sensors’ data, which were input into four different classifiers to obtain the recognition accuracy of human motion patterns. Compared with a single sensor (EMG or IMU) and single stream CNN or manual designed feature extraction methods, the feature extraction based on Dual Stream CNN-ReliefF shows better performance in terms of visualization performance and recognition accuracy. This method was used to extract features from EMG and IMU data of six subjects and input these features into four different classifiers. The motion pattern recognition accuracy of each subject under the four classifiers is above 97%, with the highest average recognition accuracy reaching 99.12%. It can be concluded that the wearable bioelectronics device and Dual Stream CNN-ReliefF feature extraction method proposed in this paper enhanced an exoskeleton’s ability to capture human movement patterns, thus providing optimal assistance to the human body at the appropriate time. Therefore, it can provide a novel approach for improving the human-machine interaction of exoskeletons.

## 1. Introduction

Exoskeleton is an intelligent wearable device, which can assist the movement of human beings’ upper or lower limbs. In recent years, it has become a research hotspot in the field of robotics. By wearing an exoskeleton, the wearer’s motor abilities and muscular endurance are enhanced, and the wearer can perform tasks that would otherwise be impossible [1]. Based on the above advantages, exoskeletons have a broad range of application prospects in medical rehabilitation, logistics, and military fields [2,3,4,5,6]. Exoskeletons rely on human-machine interaction to function efficiently. They need to recognize the current motion mode of a human body in real time so as to make corresponding motion at the appropriate time and determine the necessary assistance to the human body. Although the technology of exoskeletons has been greatly improved, there are still some deficiencies in human motion pattern recognition, wherein exoskeletons lack the ability to fully recognize the actions and intentions of human wearers. Therefore, exoskeletons cannot help the wearer at the right time, a shortcoming also highlighted in reference [7]. It is therefore necessary to study human motion pattern recognition. In this paper, the recognition of human lower limb motion pattern will be deeply studied.

In the whole process of lower limb movement pattern recognition, firstly, the human lower limb movement information should be obtained based on data acquisition equipment. Then, the obtained movement information should be extracted with features. Finally, the features are input into the classifier to identify various human movement patterns. If the motion information acquisition and feature extraction methods are not appropriate, the accuracy of human movement mode recognition will be affected. At present, a lot of work has been carried out on motion pattern recognition. Song et al. [8] used EMG sensor to collect data of five movement patterns (up the stairs, down the stairs, sit, stand and walk), and extracted frequency domain features and time domain features, including the mean frequency, median frequency, mean absolute value, wave length, slope sign change, variance, integrated, zero-crossing, and Willison amplitude. The random forest (RF) algorithm optimized by grid search method identified five motion patterns with an average recognition rate of 97.5%. Lopez-delis et al. [9] characterized knee motion patterns from EMG signals of erector spine muscles, extracted three features of mean absolute value, waveform length, and auto-regressive model and used linear discriminant analysis (LDA), K-nearest neighbor (KNN) and support vector machine (SVM) classification algorithms to identify eight human motion patterns, respectively. The recognition accuracy was greater than 95%. Xi et al. [10] extracted fifteen features (including time domain, frequency domain, time-frequency domain and entropy) and five classification algorithms for recognizing seven activities of daily life from the obtained surface EMG signals, and finally determined the optimal surface EMG features and classifiers. Peng et al. [11] used the designed plantar pressure sensing shoes to collect the plantar pressure of human body under different movement modes, and extracted five characteristics of average value, standard deviation, maximum value, minimum value, and difference deviation. The proposed KPCA-SVM classifier can recognize different motion patterns with an accuracy of 91.1%. Zhang et al. [12] collected data under dynamic (walking) and static (sitting, standing and lying) activities of the elderly by using IMU. Mean absolute value, zero-crossing, slope sign change, and waveform length were obtained, and the SVM algorithm was used to classify the above activities. A better classification effect has been achieved. Dhindsa et al. [13] obtained the EMG data of subjects during the process from sitting on a chair to standing up, extracted 15 features, and input them into the classifier after dimensionality reduction, and evaluated the performance of different classifiers. The SVMQ classifier performed best, and its recognition accuracy can reach about 92.2%. Gupta et al. [14] used acceleration sensors tied to the waist to obtain acceleration signals and extracted seven features including mean trend, windowed mean difference, detrended fluctuation analysis (DFA) coefficient, variation trend, windowed variance difference, energy uncorrelated, and maximum acceleration difference. They used Naive Bayes and KNN classifier to identify six different daily activities. The results showed that each individual activity was more than 95%. However, there are two main problems with the above work. Firstly, since the movement of human lower limbs is an extremely complex process, it needs to involve many types of data before it can be characterized comprehensively. It is difficult to fully obtain the motion information of human lower limbs by relying on a single type of sensor data. In addition, if the single sensor is affected by external uncertainty, it may lead to wrong recognition results (that is, it is difficult to meet the requirements of reliability by using a single sensor). Secondly, in the aspect of feature extraction, the time domain and frequency domain features mentioned in the above work are predefined and are manually designed based on previous research results and experience. When people use these methods to extract features, they often need to have some professional knowledge of mathematics and signal processing to select suitable features for extraction. Features extracted in this way are easily affected by the combination and number of different features, so it takes a lot of time to select the best combination and number of features, and features selected by manual experience may not be able to obtain the best classification accuracy. 

In view of the above problems, this paper is based on the advantages that EMG sensor can collect human biological information and IMU can intuitively reflect the three-dimensional kinematics information of the human body [15]. These two sensors are used to obtain more comprehensive information of human lower limb movement so as to improve the accuracy of human motion pattern recognition. Then, a feature extraction method based on Dual Stream CNN-ReliefF is proposed. It can automatically extract well differentiated fusion features from EMG and IMU data, and the extracted features have a high accuracy of motion pattern recognition. Thus, the major contributions of this paper are as follows:

Firstly, this paper presents a bioelectronic motion information acquisition device based on the fusion of EMG and IMU sensors. Compared with the information based on a single sensor, our device can collect more human lower limb motion information, which mainly consist of bioelectric information (EMG signal) and biological kinematics information (acceleration, angular velocity and angle of joint motion). It is suggested that the combination of the two kinds of information will enhance the accuracy and reliability of motion pattern recognition.

Secondly, a feature extraction method based on Dual Stream CNN-ReliefF is proposed. This method is improved on the basis of the Dual Stream CNN recognition model. The improved Dual Stream CNN-ReliefF method can automatically extract well differentiated fusion features from EMG and IMU data. The Dual Stream CNN-ReliefF feature extraction method proposed in this paper avoids the problems that the traditional manual design feature extraction involves, including a need to rely on manual experience, tedious steps, and low accuracy of feature recognition. Compared with single sensor (EMG or IMU) and Single Stream CNN or manual designed feature extraction methods, the feature extraction based on Dual Stream CNN-ReliefF shows better performance in terms of visualization performance and recognition accuracy. This method was used to extract features from EMG and IMU data of six subjects and input these features into four different classifiers. The motion pattern recognition accuracy of each subject under the four classifiers is above 97%, with the highest average recognition accuracy reaching 99.12%.

The structure of the paper is as follows: Section 2 introduces the mechanism of human lower limb movement, the wearable bioelectronics device of human lower limb movement information fusion, and the specific experimental scheme, respectively, and also introduces the feature extraction methods based on manual designs, Single Stream CNN and Dual Stream CNN-ReliefF. Section 3 presents the visual analysis of the features extracted by the three feature extraction algorithms and the accuracy analysis of motion pattern recognition under different classifiers. Section 4 presents the conclusions of our research.

## 2. Materials and Methods

### 2.1. Analysis of Human Lower Limb Movement Mechanism

Human lower limb movement is mainly completed by three parts: bone, bone connection, and skeletal muscle. Bones are the basic framework of human body; they cannot move itself and can be regarded as a rigid connecting rod. The connection between bones (that is, joints) can be regarded as motion pairs. Under the action of the nervous system, bone connection provides an external force for the joint through the contraction and relaxation of skeletal muscles in order to realize joint movement. Therefore, human lower limb movement relies on the synergistic effect between multiple joints and multiple muscle groups [16]. In order to comprehensively obtain the movement information of human lower limbs, it is necessary to obtain the movement information of each joint and muscle group of lower limbs during the movement of human body. Comprehensive movement information is helpful to improve the accuracy of human movement pattern recognition. All of the joints of the human body can only carry out rotating motion but not translational motion, which is decided by the particularity of human skeleton structure. Usually, kinematics concepts are used to represent joint motion information, such as angular velocity and angle and acceleration of joint rotation. The activity status of human muscle groups belongs to biological information. EMG signals are the superposition of action potential sequences of motor units generated during muscle contraction, which can reflect the intensity of muscle contraction. Therefore, EMG signals have been widely used in the recognition and prediction of human motion intentions [17].

### 2.2. Wearable Bioelectronics Device of Human Lower Limb Movement Information Fusion

From the above analysis of movement mechanism, it can be found that in order to comprehensively obtain the movement information of human lower limbs, a device is required to collect not only human biological information but also human kinematics information. In this paper, we proposed a set of wearable bioelectronics with human lower limb motion information acquisition device based on the EMG sensor and IMU. The device uses non-invasive EMG sensors to acquire the surface EMG signals generated by the muscles that play major roles in the movement of human lower limbs. For the collection of human kinematics information, IMU, a wearable inertial sensor, can usually be directly or indirectly placed on the human body, and can generate acceleration and rotation signals corresponding to human actions [18] so as to intuitively reflect the three-dimensional kinematics information of human lower limbs. Therefore, IMU was used to measure the angular velocity, angle and acceleration of joint rotation of human lower limbs, and these obtained data were used to represent the kinematics information. The entire wearable bioelectronics device is shown in Figure 1.

The EMG sensor (Neu Sen WM, 2000 Hz) is developed by China Changzhou Neuracle Technology Co., Ltd. (Changzhou, China). The product can simultaneously collect four channels of human surface EMG signals, and has the advantages of signal stability and high shielding. IMU (BWT901CL, 200 Hz) adopts micro-electro-mechanical system (MEMS) technology based high performance three-dimensional motion posture measurement system developed by China Shenzhen Wit-Motion Technology Co., Ltd. (Shenzhen, China). It adopts advanced filtering technology, which can effectively reduce measurement noise and improve measurement accuracy. It has a built-in three-axis accelerometer and three-axis gyroscope, which can collect the acceleration, angular velocity, and angle of object rotation, respectively. The basic principle of the wearable bioelectronics device can be described as follows: three-dimensional motion information of human lower limbs is obtained through IMU, and IMU returns acceleration, angular velocity and angle signals of three axes at a sampling rate of 200 Hz, thus forming original IMU data. Then, the original IMU data collected by the Bluetooth receiver is transmitted to the PC for storage. The EMG sensor returns four channels of human lower limb EMG signals at a sampling rate of 2000 Hz and transmits them to the amplifier. Since the human EMG signals are weak, the amplifier needs to amplify the signals. Then, the signal is sent wirelessly to the multi parameter synchronizer to form the initial raw EMG signal. Finally, the data will be transmitted to the PC for storage through a wireless mode.

### 2.3. Installation of the EMG and IMU Sensors

To obtain a human EMG signal by an EMG sensor, first of all it is necessary to determine the attachment point of the electrode in the EMG sensor in the human lower limb muscles; a reasonable electrode attachment point can better extract a human EMG signal. By consulting the biomedical experts from Aerospace Central Hospital (Asch, Beijing, China), and referring to relevant references [19,20], we learned that some muscles of the human lower limbs play a major role in the process of exercise, as shown in the following Table 1:

In the course of our experiment, we found that some muscles in lower limbs are not convenient for the posting of EMG electrodes. Meanwhile, the EMG signals of some muscles are relatively weak, which is not conducive to the analysis of human motion patterns. Based on our experimental findings and the suggestions of biomedical experts, we finally selected four muscles (rectus femoris (RF), tibialis anterior (TA), biceps femoris (BF) and gastrocnemius (GA)) as the attachment points of EMG electrodes.

According to the IMU installation, we choose to place the IMU on the thigh, calf, and heel to obtain the kinematic information of human lower limbs. Then, we found that although all IMUs have been fixed with bandages. The IMU installed on the thigh and calf move up and down to a certain extent during human movement, which will affect the stability and accuracy of data acquisition, while the IMU installed on the heel shows better stability. Based on the IMU data collected in the thigh, calf, and heel during the experiment, it is found that there exist many burrs and drifts in the kinematic image of IMU installed on the thigh and calf. However, the kinematic information obtained by IMU installed on the heel shows stability, regularity, and smoothness. To sum up, we choose to install it on the heel to measure the acceleration, angular velocity, and angle of the ankle joint rotation around the Y-axis of IMU (the X-axis is perpendicular to the ground upward, Y-axis parallel to the ground and faces to the right, and Z-axis opposite to the forward direction of human body). Since the motion amplitude of the human body in the X and Z axes of IMU is small and the motion of the exoskeleton in these two directions is generally passive, the motion in the X and Z axes will be ignored in this paper.

Before the motion information collection test, in order to reduce the influence of environmental conditions (such as humidity, electro-magnetic interference, human subject’s conditions, etc.) on EMG and IMU information, we took the following series of measurements:(1)The experiment of human motion information collection is carried out in the laboratory, and the indoor temperature and humidity remain relatively constant.(2)We removed the hair from the tested muscle, and wiped the skin with alcohol. Then, we used conductive paste to reduce the interference of skin on EMG signals and improve the conductivity and stability of EMG electrodes.(3)The backing material of the EMG electrode used is non-woven fabric, which has better air permeability. Therefore, it is suitable for long-term skin use and can reduce the interference of sweat and other factors on EMG signals.(4)The selected IMU and EMG sensors are transmitted wirelessly, which avoids the interference to the normal movement of the human body during data acquisition. The IMU uses the self-developed and improved Kalman filter fusion algorithm to solve the triaxial acceleration, angular velocity, and angle data, which avoids the situation that IMU is vulnerable to electro-magnetic interference and drift under dynamic conditions. The accuracy of acceleration, angular velocity and angle data output under dynamic conditions can reach 0.01 g, 0.05°/s and 0.03°, respectively. The EMG sensor has ultra-low input noise (<0.7 uVpp) and ultra-high input impedance (>1 GOhm), which can ensure high-quality EMG signals.(5)Based on the IMU and EMG data obtained under the above conditions, we also performed filtering (see Section 2.4 for details) to eliminate noise and interference. Through the above methods, we ensure the reliability and authenticity of the collected data.

After the placement of the electrodes and IMU, the subjects will perform some simple lower limb movements to test whether the EMG and IMU data are clear or whether there is obvious noise. The attachment points of EMG electrodes on human lower limb muscles and IMU placement are shown in Figure 2.

### 2.4. Lower Limb Movement Information Collection Scheme

This paper will consider four daily activities of the human body, including running, level ground walking, stair ascent, and ramp ascent. These daily activities can be easily carried out in the laboratory and no additional equipment is required. The four motion modes are illustrated in Figure 3.

The subjects collected the motion data of running, level ground walking, and ramp ascent on the treadmill (STAR TRAC, 10-TRx, Core health and fitness group in Vancouver, Washington, DC, USA) and stair ascent on the climbing machine (RISING, LMX-1100, Nantong, China). When we start walking or stop walking, we should stay still for 10 s so that we can better distinguish the two different states of static and walking when analyzing data. Four groups of data were collected for each motion pattern, and the duration of each group was 5 min. In order to minimize the impact of muscle fatigue on data collection, 50 min of rest is required after each motion mode. The participants included six healthy subjects, all of whom had given informed consent before participating in this study. After data collection, we use cubic spline interpolation to increase the sampling frequency of IMU from 200 Hz to 2000 Hz, which ensure that the sampling frequencies of EMG (2000 Hz) and IMU are synchronous and accurate. In addition, noise will be mixed in the process of data collection, and too much noise will adversely affect the accuracy of recognition. Therefore, denoising is necessary. Compared with other denoising methods, Butterworth can maintain good characteristics in its passband and stopband, which helps to retain useful information in the process of EMG and IMU signal denoising to obtain less signal distortion and noise. The noise of EMG signal is mainly caused by poor contact between electrode and body surface. Since the spectrum of EMG signals is mainly distributed in the range of 20 to 500 Hz [21], we use a Butterworth filter to filter out all noises outside this range. In addition, we use Butterworth low-pass filter to filter IMU data at a cut-off frequency of 10 Hz [22]. Some filtered data are shown in Figure 4a–d.

Although the quality of the measurements and the filtering algorithm have been implemented, the repeatability and reproducibility of the collected data should be verified. According to the four activity patterns of the lower limbs, including running, level ground walking, stair ascent and ramp ascent, we choose the running pattern as the analytic target, which impact the sensors data more than the others due to the large and rapid motion. Figure 4 showed parts of the IMU and EMG data from the first and second groups of running mode tests. The detailed testing method and results are as follows: (1)According to the data repeatability, the IMU and EMG sensors were fixed on the lower limbs under the same conditions in one test. Then, the data illustrate the periodical variations (partial data of IMU and EMG sensors are shown in Figure 4a,b). Therefore, the similarity coefficients between the five random cycles test data of each IMU signal were calculated by using the Pearson correlation, which were higher than 0.87. Next, the fluctuation of the average value of iEMG (integrated EMG) showed a slight oscillation, which the average value and standard deviation of RF, TA, BF, and GA are 188.609 ± 3.109, 127.313 ± 2.050, 262.253 ± 3.948 and 459.762 ± 3.274, respectively. To sum up, the results indicate that the data of five random cycles with each signal presented preferable similarity and repeatability.(2)According to the reproducibility of the data, another three group tests with the running motion mode were carried out under the same position, same sensor, and same nominal program (partial data of IMU and EMG sensors are shown in Figure 4c,d). Then, a similar approach was also adopted to validate the reproducibility with the data under the same sample periods in the four groups. Specifically, the similarity coefficients were calculated for IMU data, and the average value of iEMG were calculated for EMG data intercepted by each group of tests. The conclusions showed the similarity coefficients among each group of IMU signal data intercepted were higher than 0.85, and the average value and standard deviation of RF, TA, BF, and GA are 188.609 ± 3.109, 127.313 ± 2.050, 262.253 ± 3.948 and 459.762 ± 3.274, respectively. To sum up, it is indicated that the reproducibility among each group of data under the different tests can be demonstrated.

In summary, the collected IMU and EMG data showed available repeatability and reproducibility based on the above analysis, which can be input into the feature extraction.

### 2.5. Feature Extraction

#### 2.5.1. Extracting Features Based on Traditional Manual Design

Feature extraction plays an important part in pattern recognition, and the extraction of appropriate features will have a higher resolution, which will have an impact on the recognition accuracy [23]. The original human motion data extracted through the experiment can not be directly applied to the classification algorithm due to the large amount of data and poor performance. Therefore, it is necessary to extract the features of the original data. The processed data is more expressive and more conducive to the identification of the algorithm. Traditional feature extraction methods based on manual design tend to rely too much on manual experience and the combination and number of different features will affect the accuracy of classification. We will calculate the time and frequency domain features commonly used by IMU and EMG signals in motion pattern recognition [24,25,26], in order to compare with the feature extraction method based on Dual Stream CNN-ReliefF proposed in this paper. The specific feature expressions of IMU and EMG signals are shown in Table 2 and Table 3:

For IMU data, we calculate four time domain features (MAV, AVR, RMS, ZC) and two frequency domain features (MDF, MNP). For EMG data, we calculate six time domain features (MAV, AVR, RMS, WAMP, ZC, WL) and two frequency domain features (MDF, MNP).

#### 2.5.2. Feature Extraction Based on Dual Stream CNN-ReliefF

In recent years, CNN, an artificial intelligence algorithm, has become a hot research topic in many fields, especially in image classification. Traditional CNN classification is generally based on the structure of single stream CNN, whose model structure mainly includes the convolution layer (Conv), pooling layer (Pool), full connection layer (FC) and output layer, as shown in Figure 5. The convolution layer performs convolution operation on the input image, extracts the image features, and then outputs the convolution results by forming feature mapping through activation function. The Pooling layer is also called the down-sampling layer, whose purpose is to compress the feature graph obtained by Convolution layer, so as to optimize the computational efficiency of the model. The full connection layer maps the distributed feature representation obtained through the convolution layer and pooling layer to the sample tag space. Finally, the softmax function is applied in the output layer to export the classification results, and the error between the results and the real classification results is calculated and the weight and paranoid value are updated by back propagation. 

The dual Stream CNN recognition model is improved based on Single Stream CNN and is widely used in video action recognition [27]. The main idea is to send RGB images (representing spatial information) and optical flow images (representing temporal information) to two neural networks for feature extraction, fuse the extracted features, and finally output the classification results. Based on the above ideas, the authors designed a Dual Stream CNN feature extractor based on multiple timing information fusion. The basic principle of the extractor can be specifically described as follows: First, a complete Dual Stream CNN recognition model is built (whose structure is shown in the blue-line box in Figure 6). IMU and EMG signals were input into the model to carry out convolution and pooling operations, respectively, and two eigenvalues were obtained. Then, the two eigenvalues are fused at the convergence layer and calculated at the full connection layer. Finally, the fusion features after operation are transferred to the output layer to export the classification results. In this process, the parameters of each layer in the model are determined and retained through continuous iteration and updating of weights and deviations by back propagation algorithm. According to the loss rate and accuracy rate, the training state of the current model is judged and the excess number is adjusted in time. Finally, the batch size of the training model was set as 64, the learning rate was 0.001, and the number of iterations was 200. Use ReLu function as activation function of each convolution layer. The cross-entropy loss function is used to measure the gap between the predicted result and the actual result. The above process is the training process of Dual Stream CNN recognition model. Then, the output layer in the trained Dual Stream CNN recognition model is removed, and the remaining network structure is used to extract data features (that is, a feature extractor only used for feature extraction is formed; its structure is shown in the yellow-line box in Figure 6). In this way, the feature extractor can be used to extract fusion features from the data set composed of IMU and EMG. In this process, the feature extractor can perform in-depth feature re-mining for the two different time series information and, finally, extract well differentiated fusion features. However, the features extracted by Dual Stream CNN model still have too high data dimensions. For motion pattern recognition, when the feature dimension of the data sample is high, two aspects will be affected. First, the feature dimension is high, which will cause redundancy among features, affecting the recognition accuracy. Second, the increase of feature dimension will greatly prolong the time of model training. Therefore, it is necessary to reduce the dimension of extracted features. ReliefF is a heuristic search filter method generated by Kononenko [28] based on Kira’s work, which can process incomplete data and solve noisy multi-classification and regression problems [29,30]. At the same time, ReliefF provides feature selection with high search accuracy and efficiency. Therefore, the ReliefF feature selection method was introduced to eliminate redundant features in the initial feature set extracted by Dual Stream CNN so as to reduce the dimension of features, thus improving the classification accuracy of the model and reducing the training time of the model. The processing flow of ReliefF algorithm for feature selection is as follows:(1)First, sample *S* is randomly selected from the initial feature set.(2)From the samples with the same label as the sample, samples k are determined according to the nearest neighbor principle, constituting the sample subset H.(3)According to the nearest neighbor principle, samples k with different labels from the sample are successively determined to form the sample subset M.(4)Calculate the weight W(N) of each feature N.
(1)$$W\left(N\right)=W\left(N\right)-\frac{1}{mk}\overset{k}{\underset{i=1}{\sum }}diff\left(N,S,H_{i}\right)+\frac{1}{mk}\underset{C\notin class\left(C\right)}{\sum }\left[\frac{P\left(C\right)}{1-P\left(class\left(S\right)\right)}\overset{k}{\underset{j=1}{\sum }}diff\left(N,S,M_{j}\left(C\right)\right)\right]$$
where P(C) represents the proportion of samples with category C in the training set, and P(class(S)) represents the proportion of samples with the same category as the sample S. if N is continuous,
(2)$$diff\left(N,S_{1},S_{2}\right)=\frac{\left|S_{1}\left(N\right)-S_{2}\left(N\right)\right|}{\max\left(N\right)-\min\left(N\right)}$$
if N is discrete,
(3)$$diff\left(N,S_{1},S_{2}\right)=\{\begin{array}{c} 0,S_{1}\left(N\right)=S_{2}\left(N\right)\\ 1,S_{1}\left(N\right)\neq S_{2}\left(N\right) \end{array}$$
where $S_{1}$ and $S_{2}\;$represent the value of feature N on sample $S_{1}$ and $S_{2}$, respectively.(5)The above process is iterated m times in order to select m random samples. Finally, filter the features whose weight corresponding to the calculated features is less than the set threshold.

The above is the working principle of the Dual Stream CNN-ReliefF feature extraction model proposed in this paper. The whole process of the Dual Stream CNN-ReliefF feature extraction method is shown in Figure 6.

The specific feature extraction and fusion process is as follows:(1)The data set from IMU sensors is divided into the training set, verification set and test set according to the ratio of 7:1:2, and the data set of EMG sensors is also divided according to the above method. In order to facilitate the input of IMU and EMG data into the model, the input size of IMU data is set as 3 × 600, where 3 is the number of channels (acceleration, angular velocity, angle), and 600 is the length of input data. Then, set the input size of EMG data as 4 × 600, where 4 is the number of channels (RF, TA, BF, GA), and 600 is the length of input data.(2)Build a complete Dual Stream CNN recognition model. The purpose of inputting the training sets of IMU and EMG into the model is to train the model. In the whole process of model training, the convolution kernel size of EMG data twice convolution operation is set to 1 × 101 and 3 × 11. The convolution kernel depth is 16 and 8, respectively. The pool kernel size of the two pooling operations is set to 1 × 10 and 2 × 2, respectively. Compared with EMG data, IMU data is only different from EMG data in the convolution kernel setting in the second convolution process, which size is set to 2 × 11, and the other parameters are the same setting. After two convolution and pooling operations, the two types of training sets are expanded into 1 × 160 eigenvectors, respectively. Then, the two feature vectors are fused and operated in the convergence layer and the full connection layer to generate 1 × 320 and 1 × 32 feature vectors, respectively. Finally, the feature vector is input into the Output layer to export the classification results. After the above steps, the final required features can be formed, and the formed features will be transmitted to the classifier for motion pattern recognition. After the above steps, the complete Dual Stream CNN recognition model training is completed.(3)The training set and test set of IMU and EMG are input into the trained Dual Stream CNN recognition model, and the features before the full connection layer are reserved, which are the fusion features of the training set and test set respectively. Finally, ReliefF algorithm is used to filter the above fusion features with weights lower than 0.1.

After completing the above steps, the fusion features of training set and test set extracted by the Dual Stream CNN-ReliefF model can be obtained, respectively, and the extracted features will be input into different classifiers for motion pattern recognition.

### 2.6. Classifier

After feature extraction, another key point is to select the classifier. Since when different feature vectors and classifiers are combined, the accuracy of recognition will be different [31,32]. In this paper, we choose the commonly used classifiers in the field of motion pattern recognition, which are SVM, KNN, Decision Tree (DT) and RF [33,34,35]. In Section 3.2, we input the features extracted based on Dual Stream CNN-ReliefF, Single Stream CNN and manual design methods into the four classifiers, respectively, for comparative analysis of recognition accuracy.

## 3. Results and Discussions

### 3.1. Visual Analysis of Extracted Features Based on Different Methods and Different Types of Sensor Data

In order to verify the feature extraction capability of the Dual Stream CNN-ReliefF method proposed in this paper, it was used for feature extraction of fused EMG and IMU data, while Single Stream CNN was used for feature extraction of single EMG, single IMU, and fusion data, respectively. The extracted features are visualized in 3D images. The feature visualization images extracted based on Dual Stream CNN-ReliefF are shown in Figure 7a, and the feature visualization images extracted based on Single Stream CNN are shown in Figure 7b–d. Then, time and frequency domain features were extracted from single EMG, single IMU, and their fusion data by manual design method. Some features with better effects were visualized as shown in Figure 7e,f, respectively. 

As can be seen from Figure 7, the feature aggregation of the same motion mode and the feature differentiation of different motion modes extracted by Dual Stream CNN-ReliefF method are obviously better than those extracted by Single Stream CNN and manual design. The features extracted by Single Stream CNN and manual design have partial aliasing and poor feature aggregation. In addition, it can be seen that the features of the motion modes extracted by Dual Stream CNN-ReliefF and Single Stream CNN based on the data fusion are better than the features extracted by single EMG and single IMU data. In conclusion, we can verify that the proposed Dual Stream CNN-ReliefF method can effectively extract the features of each motion mode from the original signals. Furthermore, it can effectively characterize the four motion modes at the feature visualization performance. The visual performance of the extracted features based on the fusion of EMG and IMU data is better than that of the extracted features based on EMG and IMU data alone.

### 3.2. Recognition Accuracy Analysis of Extracted Features Based on Different Methods and Different Types of Sensor Data

In order to compare the difference in feature extraction ability among Single Stream CNN, manual design and Dual Stream CNN-ReliefF. The Dual Stream CNN-ReliefF method is used to extract features from the fused EMG and IMU data. Then these features are input into SVM, KNN, DT, and RF classifiers based on five-fold cross-validation to obtain the accuracy of motion pattern recognition. Meanwhile, the Single Stream CNN and manual design are used to extract features from a single EMG signal, a single IMU, and the fusion of the two data. Then, these features are also input into SVM, KNN, DT and RF classifiers based on five-fold cross-validation to obtain the accuracy of motion pattern recognition.

The recognition accuracy of feature extraction based on Dual Stream CNN-ReliefF and Single Stream CNN is shown in Figure 8a–d, and the recognition accuracy of feature extraction based on manual design is shown in Figure 8e–h. In addition, in the abscissa of Figure 8a–d, ISL, ESL, and IESL represent feature extraction of single IMU data, EMG data and fusion of two kinds of data based on Single Stream CNN algorithm, respectively. IEDL means feature extraction of fused EMG and IMU data based on Dual Stream CNN-ReliefF. In Figure 8e–h abscissa, ITD, IFD and ITF represent the time and frequency features extracted based on single IMU data, as well as the combination of time domain features and frequency domain features, respectively. ETD, EFD, and ETF represent the time and frequency features extracted based on single EMG data, as well as the combination of time domain features and frequency domain features, respectively. "+" indicates the fusion of time domain or frequency domain features extracted from EMG or IMU data.

As can be seen from Figure 8, the recognition accuracy of the features extracted by Dual Stream CNN-ReliefF is higher than that extracted by Single Stream CNN and manual design methods in the four classification algorithms, furthermore, the recognition accuracy of the four classification algorithms reaches more than 97%, and the best recognition accuracy can reach 99.25%. The feature extraction method based on manual design not only has tedious steps, but also the combination of different features will affect the recognition accuracy. Therefore, researchers need to rely on manual experience to pick out feature combinations with high recognition accuracy. However, relying only on manual experience is not always able to select the best combination of features, so the best recognition accuracy may not be obtained. In this paper, the best recognition accuracy is only 95.51% by selecting different feature combinations. In addition, as can be seen from Figure 8, the data based on EMG and IMU signals fusion in four kinds of classification algorithm accuracy are higher than single sensor (EMG or IMU) data. This indicates that the biological information and kinematic information brought by EMG and IMU signals can better represent the motion status of human lower limbs, thus improving the accuracy of human motion pattern recognition, indicating that the data fusion of the two sensors is effective.

### 3.3. Discussion

In order to further verify the generalization ability of the features extracted by the Dual Stream CNN-ReliefF method for motion pattern recognition of different subjects. We used the Dual Stream CNN-ReliefF method for feature extraction of EMG and IMU data from six healthy subjects. The features of each subject are input into SVM, KNN, DT, and RF classifiers based on five-fold cross-validation to obtain the accuracy of motion pattern recognition. The specific results are shown in Table 4.

As can be seen from Table 4, the motion pattern recognition accuracy of each subject under the four classifiers is above 97%, with the highest average recognition accuracy reaching 99.12%.

In addition, we compare the wearable bioelectronics device and the Dual Stream CNN-ReliefF feature extraction method proposed in this paper with other references. The specific comparison is shown in Table 5. 

Since the movement of human lower limbs is complex, which involves different kinds of information type. Compared with Re [8,11,13,14], the information obtained by a single sensor is insufficient to characterize the motion of human lower limbs, which will affect the recognition accuracy of human motion patterns. The wearable bioelectronics device proposed in this paper can collect more information of human lower limb motion, namely bioelectric information (EMG) and biological kinematics information (acceleration, angular velocity and angle of joint motion). It is suggested that the combination of these two kinds of information is more helpful to improve the accuracy of motion pattern recognition. In addition, Refs. [8,11,13,14] adopts the traditional manual design method to extract features. The steps of extracting features based on this method are not only tedious, but also the features selected by relying on human experience may not obtain best recognition accuracy when they are input into the classifier. In contrast, the Dual Stream CNN-ReliefF method proposed in this paper can automatically extract well differentiated fusion features from EMG and IMU data. The proposed method was used to extract features from six subjects, and these features were input into four different classifiers (SVM, KNN, DT, RF). The motion pattern average recognition accuracy obtained through five-fold cross-validation was up to 99.12%. However, compared with other references, the inadequacy of this paper is the lack of more research on the types of human lower motion pattern, such as jumping, squatting and so on. 

Therefore, the wearable bioelectronics device based on EMG and IMU with the Dual Stream CNN-ReliefF feature extraction method proposed in this paper can improve the accuracy of human motion pattern recognition, and have better generalization ability for the motion pattern recognition of different subjects.

## 4. Conclusions

This paper firstly analyzed the motion mechanism of human lower limbs, and designed a set of wearable bioelectronics device based on EMG and IMU sensors, which can obtain biological information and kinematics information of human lower limbs. Based on the information fusion, we collected the lower limb movement information of four common movement modes of human body. Then, a feature extraction method based on Dual Stream CNN-ReliefF is proposed, which can automatically extract well differentiated fusion features from EMG and IMU data. Finally, the Dual Stream CNN-ReliefF was adopted for feature extraction of the fusion data of EMG and IMU. Meanwhile, the Single Stream CNN and manual design were used for feature extraction of single EMG, single IMU, and their fusion data, respectively. The extracted features based on the above three methods are analyzed and compared in feature visualization and recognition accuracy. We concluded that the wearable bioelectronics device information fusion and Dual Stream CNN-ReliefF feature extraction method proposed in this paper enhanced an exoskeleton’s ability to capture human movement patterns, thus providing optimal assistance to the human body at the appropriate time. Therefore, it can provide a novel approach for improving the human-machine interaction of exoskeletons. In the future, on the one hand, we will study the recognition of more motion patterns of human lower limbs; on the other hand, we will conduct indepth research on the motion pattern recognition of human upper limbs, so as to provide certain direction for the motion pattern recognition of exoskeletons in human upper and lower limbs.

## Figures and Tables

**Figure 1 micromachines-13-01205-f001:**
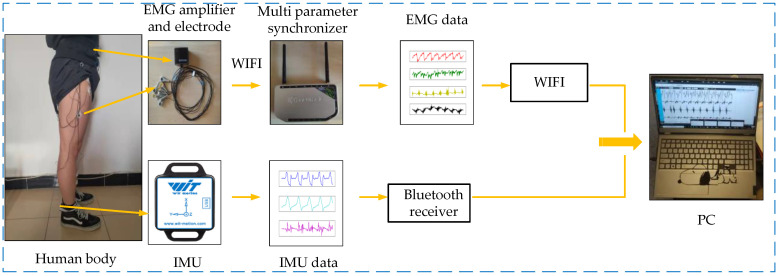
Wearable bioelectronics device of human lower limb movement information fusion.

**Figure 2 micromachines-13-01205-f002:**
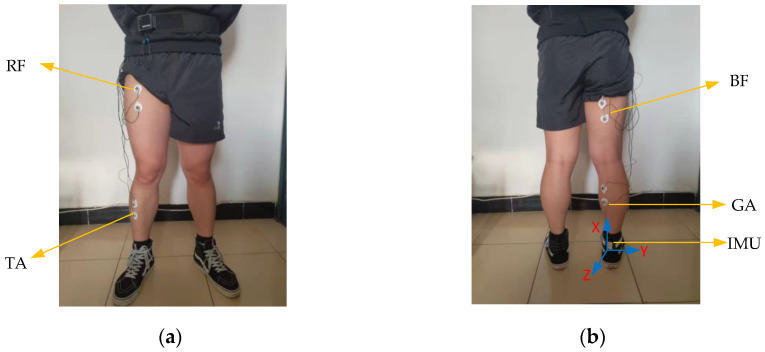
Electrode attachment point and IMU position. (**a**) Frontal side of human lower limbs; (**b**) back side of human lower limbs.

**Figure 3 micromachines-13-01205-f003:**
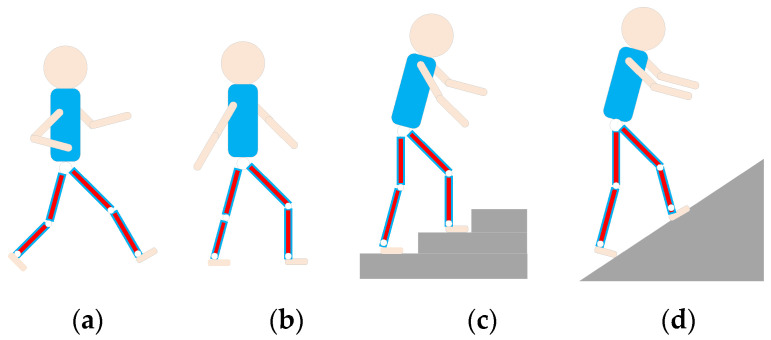
Four daily patterns of human movement. (**a**–**d**) represent running, level ground walking, stair ascent and ramp ascent.

**Figure 4 micromachines-13-01205-f004:**
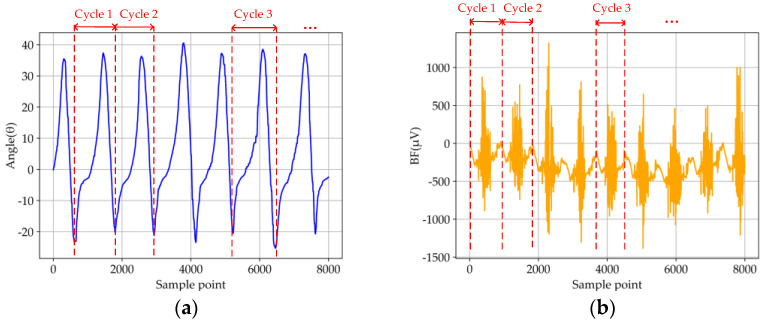
Parts of the test data obtained from the first and second groups of running mode tests. (**a**) Angle signal by IMU in the first group test; (**b**) EMG signal of BF in the first group test; (**c**) angle signal by IMU in the second group test; (**d**) EMG signal of BF in the second group test.

**Figure 5 micromachines-13-01205-f005:**
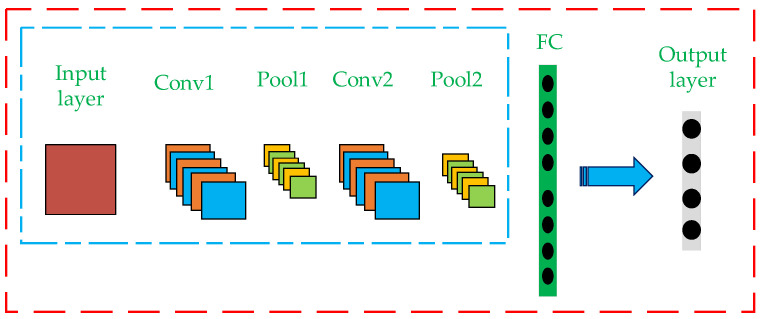
Single Stream CNN algorithm structure diagram.

**Figure 6 micromachines-13-01205-f006:**
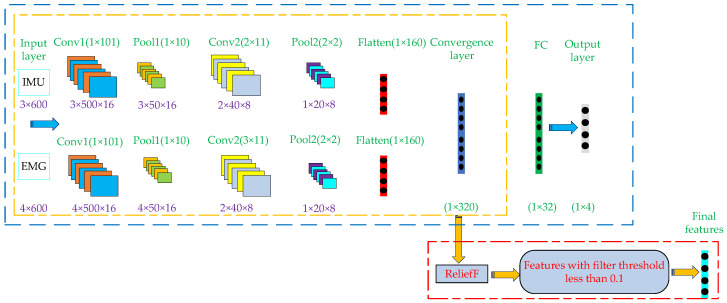
The whole process of feature extraction based on Dual Stream CNN-ReliefF.

**Figure 7 micromachines-13-01205-f007:**
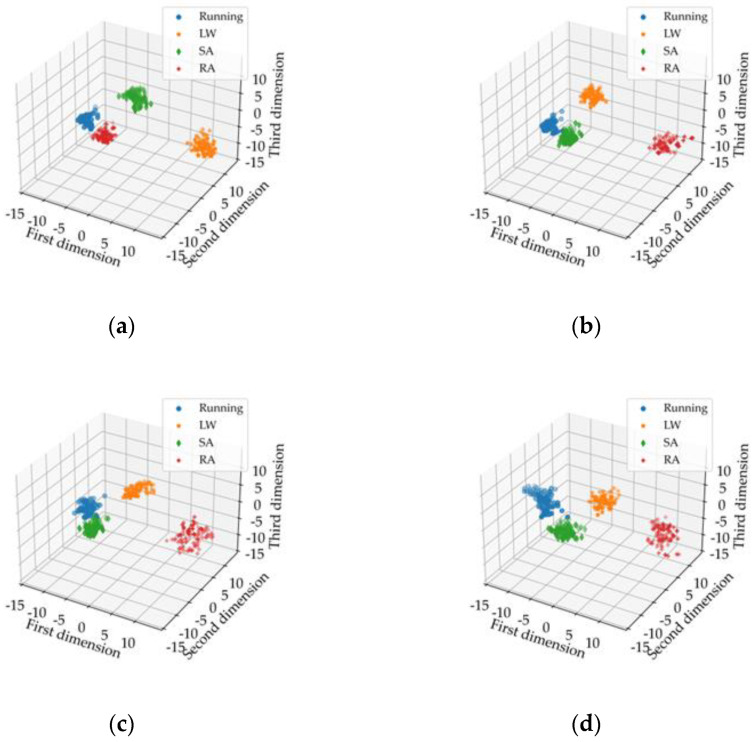
Visualization of feature extraction based on different methods and different types of sensor data. LW (level ground walking), SA (stair ascent), RA (ramp ascent). (**a**) represent the features extracted by using Dual Stream CNN-ReliefF based on the fusion of EMG and IMU data; (**b**–**d**) represent the features extracted by using Single Stream CNN based on a single EMG, a single IMU and the fusion of the two data, respectively; (**e**,**f**) represent the features with better visual effects extracted by manual design based on a single EMG, a single IMU and the fusion of the two data, respectively.

**Figure 8 micromachines-13-01205-f008:**
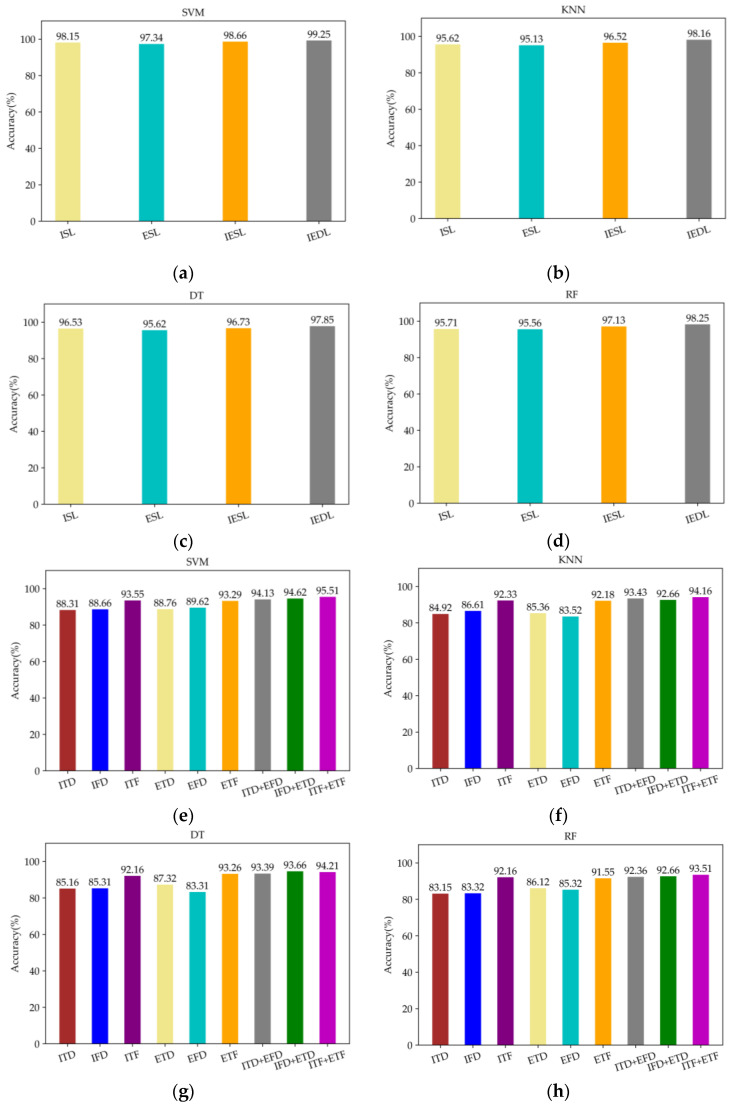
Recognition accuracy analysis of extracted features based on different methods and different types of sensor data. (**a**–**d**) represent the recognition accuracy of features extracted based on Dual Stream CNN-ReliefF and Single Stream CNN under SVM, KNN, DT and RF classifiers, respectively; (**e**–**h**) represent the recognition accuracy of features extracted based on manual design under SVM, KNN, DT and RF classifiers, respectively.

**Table 1 micromachines-13-01205-t001:** Muscles and functions that play a major role in lower limb movement.

Muscle	Motor Function
rectus femoris (RF)	stretch the calf, bend the thigh
vastus lateralis (VL)	stretch the calf
gastrocnemius (GA)	bend the calf, lift the heel, fix the knee, balance the body
musculus peroneus longus (ML)	foot valgus, plantarflexion
tibialis anterior (TA)	dorsiflexion, varus, adduction
biceps femoris (BF)	bend the calf, stretch the thigh, rotate the calf outward
soleus (SL)	bend the calf, lift the heel, fix the knee, balance the body
semitendinosus (SD)	stretch the thigh, bend the calf, rotate the thigh inside

**Table 2 micromachines-13-01205-t002:** Time and frequency domain features expression of IMU.

Feature	Mathematical Definition
Mean Absolute Value (MAV)	$\frac{1}{N}\overset{N}{\underset{i=1}{\sum }}|x_{i}|$
Variance (VAR)	$\frac{1}{N-1}\overset{N}{\underset{i=1}{\sum }}x_{i}^{2}$
Root Mean Square (RMS)	$\mathrm{RMS}=\sqrt{\frac{1}{N}\overset{N}{\underset{i=1}{\sum }}x_{i}^{2}}$
Zero Crossing (ZC)	$\{\begin{array}{c} \mathrm{ZC}=\overset{N}{\underset{i=1}{\sum }}z_{i}\times zc_{i}\\ zc_{i}=\mathrm{sgn}\left(-x_{i}x_{i+1}\right)\\ z_{i}=\{\begin{array}{c} 1,\left|x_{i}-x_{i+1}\right|>\delta_{z}\\ 0,else \end{array} \end{array}$
Median Frequency (MDF)	$\overset{\mathrm{MDF}}{\underset{i=f_{min}}{\sum }}p_{i}=\overset{f_{max}}{\underset{i=\mathrm{MDF}}{\sum }}p_{i}$
Mean Power (MNP)	$\mathrm{MNP}=\overset{M}{\underset{j=1}{\sum }}p_{j}/M$

**Table 3 micromachines-13-01205-t003:** Time and frequency domain features expression of EMG.

Feature	Mathematical Definition
Mean Absolute Value (MAV)	$\frac{1}{N}\overset{N}{\underset{i=1}{\sum }}|x_{i}|$
Variance (VAR)	$\frac{1}{N-1}\overset{N}{\underset{i=1}{\sum }}x_{i}^{2}$
Root Mean Square (RMS)	$\mathrm{RMS}=\sqrt{\frac{1}{N}\overset{N}{\underset{i=1}{\sum }}x_{i}^{2}}$
Wilson Amplitude (WAMP)	$\mathrm{WAMP}=\overset{N-1}{\underset{i=1}{\sum }}\left[f\left(\left|x_{n}-x_{n+1}\right|\right)\right]$ $f\left(x\right)=\{\begin{array}{c} 1,if\;x\geq threshold\\ 0,\;\;\;\;\;\;\;\;\;\;\;\;\;\;\;otherwise \end{array}$
Zero Crossing (ZC)	$\begin{array}{c} \mathrm{ZC}=\overset{N}{\underset{i=1}{\sum }}z_{i}\times zc_{i}\\ zc_{i}=\mathrm{sgn}\left(-x_{i}x_{i+1}\right)\\ z_{i}=\{\begin{array}{c} 1,\left|x_{i}-x_{i+1}\right|>\delta_{z}\\ 0,else \end{array} \end{array}$
Waveform Length (WL)	$\overset{N-1}{\underset{i=1}{\sum }}\left|x_{i+1}-x_{i}\right|$
Median Frequency (MDF)	$\overset{\mathrm{MDF}}{\underset{i=f_{min}}{\sum }}p_{i}=\overset{f_{max}}{\underset{i=\mathrm{MDF}}{\sum }}p_{i}$
Mean power (MNP)	$\mathrm{MNP}=\overset{M}{\underset{j=1}{\sum }}p_{j}/M$

**Table 4 micromachines-13-01205-t004:** The accuracy of motion pattern recognition of different subjects under five-fold cross-validation.

Subject	Age	Height [cm]	SVM	KNN	DT	RF	Average Accuracy
S1	23	167	99.25%	98.16%	97.85%	98.25%	98.37%
S2	24	170	99.46%	99.15%	98.36%	98.63%	98.90%
S3	26	165	99.69%	97.68%	99.13%	97.66%	98.54%
S4	28	172	99.13%	99.38%	97.46%	98.17%	98.53%
S5	29	176	99.73%	99.34%	98.25%	99.16%	99.12%
S6	30	173	99.67%	98.61%	98.74%	99.31%	99.08%

**Table 5 micromachines-13-01205-t005:** Comparison of the methods and experimental results.

Research	Sensor	Feature	Motion Pattern	Accuracy
Song [8]	EMG	9 time-frequency features	5 motion patterns	97.5%
Peng [11]	Plantar pressure	5 time-frequency features	5 motion patterns	91.1%
Dhindsa [13]	EMG	15 time-frequency features	2 motion patterns	92.2%
Gupta [14]	Acceleration	7 time-frequency features	6 motion patterns	>95%
This work	EMG + IMU	Dual Stream CNN-ReliefF	4 motion patterns	99.12%

## Data Availability

The data presented in this study are available on request from the corresponding author on reasonable request.

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
