# Peer review of "Human Motion Pattern Recognition and Feature Extraction: An Approach Using Multi-Information Fusion"

_micromachines, 2022, doi:10.3390/mi13081205_

Round 1
Reviewer 1 Report
The work presented in this manuscript is mainly on using dual stream convolutional neural network ( aka CNN-ReliefF) to identify human lower limb motion based on EMG and IMU signal outputs. The work may be possible to be included int his journal should the following points be clearly improved.
(1) The subject of the work is interesting. However, the novelty of the approaches/methods used in the work presented in this manuscript is insufficient. It is not clear to this reviewer what new knowledgement has been generated or "significant" improvement has been demonstrated by the proposed methods.
(2) It is well known that signals of EMG and IMU can be significantly affected by the ways the sensors are placed to human body as well as ambient conditions such as humidity, electro-magnetic interferenece, or even human subject's conditions. However, the work presented in this manuscirpt didn't even discuss or demonstrated methods to improve signals from the mentioned factors but rather in the use of CNN-ReliefF to process the signals. Therefore, rather than showing the authors can use the method, the approaches are not scientific rigious, which prevent the present work to be able for possible archival publication in this journal.
(3) Since the subject research area has been well studies, it is suggested that authors should compare results with the proposed CNN-ReliefF methods with other methods from the literatures to show pros and cons of the proposed method.
(4) As the sensors are placed to human body and it is apparently that locatoin of placement is gated by biomedical domain know-hows. It is suggested that the authors should clearly describe why the sensors are placed on the specific locations with the help from biomedical experts such as physicians. Without such statement, the results are not that meaningful.
Reviewer 2 Report
1. This paper only collects and fuses the information of sEMG and IMU sensors. The title is that multi-information fusion is not accurate, and it is suggested to change the title.
2. The author said that installing IMU on the heel can better represent the kinematic information. What is the basis? As far as reviewers know, IMU sensors are prone to drift during motion information collection. How can the author avoid this phenomenon and deal with the data collected by IMU to ensure the authenticity of the data?
3. The experimental method needs to be explained with more detailed information, and supplementary experiments are needed to verify the generalization ability of the method/model to different subjects' motion pattern recognition.
4. The author needs to give a detailed explanation or description of the process of information fusion by the fusion model, otherwise it will be difficult for readers to understand.
5. There is not enough details about model selection and evaluation? For example, the model's hyper-parameters. In addition, the author should cross-validate the model to adequately assess its performance. Those details are crucial to test any model, let alone black-box machine learning models. The more sensor we have, the more challenging it will be to replicate the results in real world and therefore, a thorough analysis of unseen data need to be conducted.
6. The sEMG signal and the IMU data are sampled at different frequencies, and the authors need to explain how to overcome the discrepancy and ensure synchronous data collection.
7. For pre-processing of EMG and IMU signals Butterworth filter is used. From many denoising methods, why Butterworth denoising is used, any justification.
8. The format of references needs to be uniform, such as 7, 8, 11, 17, and 20.
9. English needs to be improved
